# Inconsistent Provider Testing Practices for Congenital Cytomegalovirus: Missed Diagnoses and Missed Opportunities

**DOI:** 10.3390/ijns8040060

**Published:** 2022-11-14

**Authors:** Kate L. Wilson, Kimi Shah, Megan H. Pesch

**Affiliations:** 1Division of Neonatal-Perinatal Medicine, Department of Pediatrics, University of Michigan, Ann Arbor, MI 48109, USA; 2University of Michigan Medical School, Ann Arbor, MI 48109, USA; 3Division of Developmental and Behavioral Pediatrics, Department of Pediatrics, University of Michigan, Ann Arbor, MI 48109, USA

**Keywords:** congenital cytomegalovirus, testing, provider practice, clinical suspicion, diagnosis

## Abstract

Newborn congenital cytomegalovirus (cCMV) screening programs have been found to increase the rates of early diagnosis and treatment. In North America, newborn cCMV screening programs have not been widely implemented, leaving healthcare providers to rely on clinical suspicion alone to prompt testing. This study sought to examine healthcare providers’ cCMV testing practices at a quaternary children’s hospital. A retrospective review of the electronic health record was completed for eligible infants over a six-year period. Bivariate calculations and analyses were performed. Between 2014 and 2019, a total of 40,091 infants were cared for at the study institution, of which 178 were tested for cCMV and 10 infants were diagnosed with cCMV. Isolated small-for-gestational age was the most common indication (53/178) to prompt testing. Overall, the cCMV testing rate was 4.5 tests per 1000 infants, with a resulting diagnostic prevalence of 0.2 cases per 1000 infants, which is 15-fold lower than the expected prevalence. Providers relying on clinical suspicion alone are infrequently testing infants for cCMV, resulting in missed diagnoses and missed opportunities for treatment. Systematic cCMV screening practices may improve diagnosis, treatment, and childhood outcomes.

## 1. Introduction

Congenital cytomegalovirus (cCMV) is the most common congenital infection in the United States (US) affecting approximately 3–6 per 1000 live births [1,2]. Approximately 90% of infants with cCMV are born with asymptomatic infections, defined as not having any clinical signs on physical exam, cranial imaging or laboratory tests (e.g., thrombocytopenia or elevated liver enzymes) [3]. The remaining 10% are born with symptomatic disease, which is characterized by visible signs at birth, laboratory abnormalities and/or intracranial involvement [3]. Yet, in many infants the visible signs of symptomatic cCMV disease may be subtle, often non-specific (e.g., jaundice or small-for-gestational age), and/or may occur in isolation [4,5]. The International Congenital Cytomegalovirus Recommendations Group (ICCRG) defined disease severity based on the number of manifestations attributed to cCMV and whether they occurred in isolation or not, thus differentiating mild disease from moderate to severe symptomatic disease [3]. Other critical markers of symptomatic disease, such as laboratory abnormalities and intracranial lesions, are not always outwardly noticeable and are only identified on further work-up after clinical suspicion is raised [3,4].

As such, it is likely that many infants born with cCMV go undiagnosed when testing relies on the clinical recognition and suspicion of providers alone, although how many infants are being missed is a point of some debate. One study from British Columbia, Canada reported that only 10% of infants with symptomatic cCMV were being diagnosed, or 1% of all infants with cCMV [6], which is significantly lower than the estimated national prevalence of 3–6 cases per 1000 live births [1]. Other work using administrative claims data from US hospital discharges identified cCMV diagnosis codes in approximately 2 per 10,000 infants, presumably all coded for symptomatic infections [7]. Thus, nearly 50% of all expected cases of symptomatic cCMV, or 5% of all cases, were identified, which is 5x greater than the Canadian study [7]. As there is a greater urgency to diagnose, evaluate, and initiate treatment for infants with symptomatic cCMV, if half are already being diagnosed based on clinical symptoms alone, then the utility of a newborn screening program may not be as dramatic.. However, asymptomatic infants are likely to continue to go unrecognized and undiagnosed.

A timely diagnosis of cCMV is critical for access to early interventions, which may include antiviral medication, hearing monitoring and amplification as well as developmental therapies [3,8,9]. Given the clinical challenge of and narrow window for diagnosing cCMV, enthusiasm for neonatal cCMV screening programs has grown, but at present they are only implemented in a handful of states [10]. To date, little is known about US healthcare providers’ practices surrounding cCMV testing in infants. While neonatal cCMV screening programs have yet to be endorsed by the American Academy of Pediatrics, a better understanding of cCMV testing practices when based on clinical suspicion alone, including the indications that prompt providers to test for cCMV, may inform the need for and the content of future guidelines.

Therefore, this study sought to examine cCMV testing practices at a quaternary US children’s hospital based on clinical suspicion alone over a six-year period. Specifically, the objectives of this study were to examine: (1) the circumstances surrounding each cCMV test ordered (number tested, test result, type of testing modality, clinical unit that ordered the testing, and the indications for testing); (2) the differences in the sociodemographic and clinical characteristics of infants based on whether they were tested for cCMV and the test results; and finally, (3) the prevalence of cCMV diagnoses made in the study population as compared to the true prevalence of cCMV based of epidemiologic estimates. This study focused on infants less than or equal to 31 days old, and who were all cared for at the same hospital, which did not have a systematic cCMV screening program in place.

## 2. Methods

### 2.1. Study Overview

We performed a retrospective review of the electronic health record (EHR) for all infants cared for at a health system located in midwestern US, which included a quaternary children’s hospital attached to a women’s labor and delivery hospital over a six-year period. We identified infants who were cared for at the study institution by using an electronic cohort discovery tool, and manually reviewed the clinical medical records of all infants meeting the inclusion criteria who were also tested for CMV. The data collected included sociodemographic and clinical variables. The University of Michigan Medical School Institutional Review Board granted exemption status for this study (HUM00190522).

### 2.2. Cohort Creation

We used DataDirect [11], a self-service cohort query tool that searches for clinical data from an electronic data repository within the study institution’s EHR to identify the study cohort. We included infants ≤31 days old who received care in the health system (both inpatient and outpatient) during the study period of 1 January 2014 to 31 December 2019. The age of ≤31 days was selected as this is the upper limit for the initiation of antiviral treatment in moderately-to-severely symptomatic infants [9]. In DataDirect, we then further limited the search to those infants who had a laboratory test for CMV ordered with subsequent results. The DataDirect tool allows for tests to be selected based on keywords or the test name/abbreviation used at the study institution. We identified 98 laboratory test results and orders with either “CMV” or “cytomegalovirus” in their test name or keywords (assigned by the study institution), and after the manual review we included 46 laboratory tests associated with results for CMV in the final search. We did not include laboratory tests associated with orders only, nor did we include tests that could not be used to test for cCMV (e.g., bone marrow biopsy aspirate PCR). The laboratory tests included those seeking to isolate the virus in plasma, urine, dried blood spot, or cerebral spinal fluid (CSF) by polymerase chain reaction (PCR), or by viral culture, or CMV anti-bodies in plasma. Saliva CMV PCR testing was not available at the study institution during the study period. Please reference the Appendix A [11] for additional details including the list of all laboratory test orders and the rationale for inclusion.

We then performed a manual medical record chart review for each infant to determine if infants were tested for congenital vs. post-natal CMV infections. This differentiation was made based on a review of the provider notes, clinical reasoning behind ordering CMV testing, and if testing was obtained prior to 21 days of life. For example, some infants were tested for post-natal CMV infections as part of an organ transplant evaluation, or sepsis, whereas for others, the concern regarding cCMV was specifically mentioned in their clinical record.

Of the cohort of 40,091 infants identified, 234 were tested for any CMV (congenital or postnatal) infection, of which 178 infants met the criteria for testing for cCMV and 56 infants met the criteria for testing for postnatal CMV infections. This resulted in a total cohort of 40,091, made up of 39,913 infants not tested for cCMV, and 178 tested for cCMV. A flowchart of the creation of the cohort is shown in Figure 1.

### 2.3. Data Collection

For the study cohort of infants tested for cCMV (N = 178), we extensively reviewed the medical record of each infant from birth to the date that the CMV test was ordered. We finalized the data collection form through an iterative process described below and added additional variables (columns) throughout the process, as necessary. When we added a variable (e.g., hepatosplenomegaly was separated into two variables: hepatomegaly and splenomegaly), prior records were re-reviewed to ensure accurate data. Reliability between the study team members reviewing the records (first MP and KW, then later KS) was established by each team member independently reviewing the same randomly chosen set of 10 records, guided by a mutually agreed upon set of definitions for each variable. The reviewers then met to compare their data collected, and discussed agreements, as well as disagreements, and to add needed detail to the definitions. Disagreements were discussed and an agreement was reached. An additional 20 records were reviewed independently, after which reliability was achieved (100% agreement, Cohen’s kappa = 1.0). Thereafter, the remaining records were reviewed by a single reviewer (MP or KW, not both) over the course of eight weeks. Later during the analysis phase of the project, we added two additional binary variables: (1) elevated ALT > 100 IU/L (yes vs. no or unknown), and (2) elevated AST > 80 IU/L (yes vs. no or unknown). KS reviewed the charts of all 178 infants tested for cCMV and collected this data. Formal reliability was not calculated although 10 random participant’s charts were verified by KW after coding was completed. All collected elements and variable definitions as well as further details can be found in the Appendix A [3,12,13,14,15,16,17,18].

We reviewed the following elements of the medical record including clinical notes (including physical examination) as written by physicians, nurse practitioners, and/or physician assistants, growth measurements adjusted appropriately by gestational age at birth, laboratory tests ordered and the results, imaging studies and the final read, newborn hearing screening and the results, and/or ophthalmologic examination and the findings.

The circumstances around cCMV testing included the following variables: date of the test, specific test type ordered, test result, the clinical unit that ordered the test, and the clinical indication as documented by the provider that prompted cCMV testing. An infant met the criteria for cCMV infection if they had a documented positive urine, serum/plasma, or CSF CMV PCR, or culture performed prior to 21 days of age, or with a confirmatory dried blood spot PCR if tested ≥21 days of age if the sample was collected between 21 and 31 days of age [3]. Antibody tests not followed up by PCR testing were not considered to be sufficient to confirm a cCMV infection.

The clinical indication for cCMV testing was defined as the clinical reasoning as described by the provider(s) in the infant’s clinical medical record that raised suspicion for cCMV and prompted testing. We also reviewed the medical record for clinical findings such as signs on physical exam, laboratory tests, imaging studies, hearing screening/tests, or ophthalmologic exams [3,4]. Clinical findings (objective findings) were not assumed to be the same as clinical indications (providers’ subjective reasoning for testing); thus, these variables were collected separately.

For the larger cohort of infants not tested for CMV (n = 39,913), we extracted data from the EHR using the Oracle™ Database 12c Enterprise Edition [19], which included discrete sociodemographic and clinical data elements, as similarly explained above. The presence or absence of clinical diagnoses for this larger cohort was determined by the presence (vs. absence) of diagnoses, and/or laboratory test values above a certain threshold (Appendix A [3,12,13,14,15,16,17,18]). Birth-related elements could only be extracted from the EHR for infants born at the study institution (n = 31,402).

### 2.4. Statistical Analysis

We calculated the descriptive characteristics of the sample, as well as bivariate calculations comparing the characteristics of infants in the larger cohort, all infants tested for cCMV, infants with negative cCMV results and positive cCMV results using z-scores for the categorical variables and *t*-tests for continuous variables using Microsoft Excel. We calculated the prevalence of cCMV testing and true diagnoses. We calculated the 95% confidence intervals for the proportion of positive tests, or percent positive values, using a binomial Wilson score interval due to the small sample size in R [20]. A *p*-value of <0.05 was considered statistically significant.

## 3. Results

A flowchart of infants tested for cCMV based on the ordering unit, testing indications, and results is presented in Figure 1. Of the 178 infants tested for cCMV, 12 had positive results confirming a diagnosis of cCMV (percent positive 6.7%, 95% CI 3.9–11.4), only 10 of whom were identified based on the clinical suspicion of providers at the study institution (percent positive 5.6%, 95% CI 3.1–10.0). Two of the 12 infants diagnosed with cCMV were transferred from a referring institution’s Neonatal Intensive Care Unit (NICU) to the study institution’s NICU secondary to an escalation of acuity and/or need for sub-specialty consultation due to a suspicion for congenital infections. At the time of transfer both infants had dedicated CMV testing at the referring institution. As these results were not yet available at the time of admission, the study institution ordered additional CMV PCR testing (all tests eventually came back positive). As such, we consider these two infants as being identified by their referring institution. All 12 infants had “congenital CMV” or “congenital CMV infection” associated with International Classification of Diseases 9 or 10 codes (771.1 and/or P35.1) [21,22] in their clinical records.

Quantitative CMV PCR (urine or plasma/serum) was the most used testing modality (142/178) as depicted in Figure 2. However, early in the study period, viral culture was more commonly ordered and then it phased out to exclusive PCR testing by 2016. Antibody testing was used in five infants in 2014, and once in 2018. Of note, CMV antibody testing is not recommended for congenital CMV, as maternal antibodies are found in the neonatal circulation. Dried blood spot or CSF CMV PCR were not used to test any infant in the study sample.

Infants tested for cCMV were cared for by five different units/divisions: the NICU, Newborn Nursery, Pediatric Intensive Care Unit (PICU), Pediatric Cardiac Intensive Care Unit (PCTU) and the inpatient general care Pediatric service. Infants cared for by the NICU represented the majority (72%) of those tested for cCMV. The percent positive for tests ordered by the NICU was 2.3% (95% CI 0.8–6.7) or 0.8% (95% CI 0.14–4.4) after the removal of the two infants transferred with pending cCMV testing. In comparison, the percent positive of the tests ordered by providers in a non-ICU setting was 27.6% (95% CI 14.7–45.7) and 7.7% (95% CI 1.4–33.3) for the Newborn Nursery and Pediatric general care units, respectively.

The clinical indications documented by healthcare providers that prompted testing are displayed in Figure 3. Small-for-gestational age (SGA), defined as a birthweight for sex <10th%ile [12,13] (53/178) and intracranial abnormalities (15/178) were the most documented testing indications. A failed newborn hearing screen only prompted five providers to order cCMV testing. While SGA was the most common indication that prompted testing, it only yielded three positive cCMV diagnoses, all of which were cared for by providers in the Newborn Nursery, as illustrated in Figure 1. Prenatal concern for CMV due to positive maternal serologies was the testing indication with the highest yield for positive diagnoses (4/12).

The sociodemographic characteristics of the cohort are presented in Table 1a. Most infants tested for cCMV were male (54.5%) and of Caucasian race (66.9%); these proportions did not statistically differ from the larger cohort. Infants tested for cCMV disproportionately had public (vs. private or other) insurance as compared to those not tested for cCMV (*p* < 0.001). Infants who tested positive for cCMV had mothers with higher parity than those who tested negative or those not tested at all (*p* < 0.001).

The birth characteristics of the cohort are shown in Table 1b. Infants tested for cCMV were more likely to have been inborn (vs. at an outside hospital), born preterm (vs. term), and have a lower birthweight and head circumference percentile for gestational age, than those not tested. Infants who tested positive for cCMV (vs. negative) were less likely to be born premature and had a lower mean birthweight percentile for gestational age (17.0%ile vs. 31.5%ile, *p* < 0.0001), but there was no difference in gestational age at birth or head circumference percentile. The most common clinical findings in infants tested for cCMV were jaundice (52.2%), SGA (46.6%), microcephaly (44.9%) and thrombocytopenia (46.1%). Clinical findings or signs that are classically associated with cCMV were more common in those tested for cCMV (vs. not) (all *p* < 0.003). Infants who tested positive for cCMV (vs. tested negative), disproportionately had microcephaly, splenomegaly, petechiae, a failed newborn hearing screen, intracranial abnormalities on imaging, and/or an elevated aspartate aminotransferase (all *p* < 0.001). Of the infants diagnosed with cCMV, 2 had asymptomatic infections (no clinical signs) and 10 had symptomatic disease, based on the disease classification as defined in the consensus recommendations by the International Congenital Cytomegalovirus Recommendations Group [3].

The overall cCMV testing rate in this study was 4.5 tests per 1000 infants, with a resulting prevalence rate of diagnosed cCMV of 0.32 cases per 1000 infants cared for at the study institution. However, 2 of the 12 infants were transferred to the NICU with cCMV testing in process due to the referring provider’s clinical suspicion for congenital infection, resulting in only 10 infants with cCMV whose testing was initiated based on the study provider’s clinical suspicion. As such, the prevalence rate of diagnosed cCMV based on the clinical suspicion of providers at the study institution was 10/40,091 or 0.2 cases per 1000 infants.

Reported estimates of the prevalence of cCMV in the US range from 3 to 6 cases per 1000 live births [1,2]. Using a median estimate of 4.5 cases per 1000 live births, 180 cases of cCMV, including 18 symptomatic and 162 asymptomatic infants, would be expected among the cohort of 40,091 infants. With those estimates in mind, the rate of cCMV diagnosis in this study was 15-fold, 2-fold and 81-fold lower than the expected prevalence of all cCMV, symptomatic cCMV, and asymptomatic cCMV infections, respectively.

## 4. Discussion

This study found a low rate of cCMV testing and diagnosis in a cohort of over 40,000 infants cared for in a quaternary US children’s hospital without a systematic cCMV screening program in place. The indications that prompted testing were multifarious. While isolated SGA was the most common single indication for testing, positive maternal prenatal CMV serologies was the most common indication that was associated with a positive result. Of the infants found to have cCMV with testing prompted by positive maternal prenatal serologies, two of the four infants had no clinical signs of disease at birth and would have likely otherwise gone undiagnosed. As cCMV testing was infrequent, the diagnostic prevalence of cCMV in this cohort was low. The prevalence of cCMV in the United States is estimated to be 4.5 per 1000 live births [1]; this present study found the diagnosed prevalence to be 0.3 per 1000 live births in total (including outside hospital transfers for likely cCMV), or 0.2 per 1000 live births based on clinical suspicion of the providers at the study institution alone. Our findings of low rates of cCMV testing and diagnosis echo those of prior work [6], and it is likely that most cases of cCMV go undiagnosed when a systematic cCMV testing or screening program is not in place [6,23,24,25]. We hypothesize that the low testing rate may be due to low cCMV awareness among clinicians, and the often subtle or inapparent clinical presentation of cCMV [26,27,28].

In this study, there did not seem to be a uniform or best practice in terms of clinical indications for cCMV testing. While isolated SGA was the most documented testing indication, it did not have a high yield association with a positive cCMV result. Providers in the NICU predominantly tested infants based on this indication and yielded zero positive cases, whereas the Newborn Nursery tested far less and yielded three positive cases. While this may raise the question about the utility of testing for cCMV based on a clinical indication of SGA in the NICU, it must be noted that not all infants with SGA cared for in the NICU were tested for cCMV, and as such caution must be used in generalizing these results beyond the tested cohort. Our findings parallel those of Smiljkovic et al. [29], who reported screening indications for children diagnosed with cCMV by retrospective review of the EHR in Quebec, Canada. Smiljkovic et al. reported infant clinical, laboratory or imaging findings concerning for cCMV, which they termed “symptomatic infant” as being the most common cCMV testing indication (51%) among 47 infants with cCMV, followed by maternal serologies [29]. Three quarters of the infants with cCMV in our study were tested due to clinical, laboratory or imaging findings, therefore 75% of the primary screening indications were due to being a “symptomatic infant”, and 25% were due to maternal serologies. Although the definitions used for specific clinical findings varied between studies, our findings regarding the proportion of infants with cCMV that had specific clinical signs were largely similar to prior studies using review of the clinical medical record: evidence of CNS involvement (75% vs. 52–58%), intrauterine growth restriction/SGA (33% vs. 38–48%) and thrombocytopenia (50% vs. 36%) [29,30,31]. The proportion of infants with cCMV with these clinical signs in our study is higher than estimates based on administrative claims data. For example, microcephaly was found in 58% of infants with cCMV in our study vs. 6–11% in two studies of administrative data reliant on diagnostic codes [14,32]. It is likely that administrative claims analyses underestimate the prevalence of clinical signs, as they are reliant on the provider documenting the sign as a separate diagnosis, rather than in the clinical note alone [33].

Prenatal maternal serologies accounted for 25% of the indications for cCMV testing which were associated with a positive cCMV diagnosis. This may indicate that prenatal serologies could be important for cCMV detection at birth. At present, routine prenatal CMV serologic testing is not recommended by US professional organizations due to the challenge of interpreting results [34]. In other countries, especially where seroprevalence is low, prenatal CMV serologic testing is common practice in order to monitor for maternal seroconversion throughout the pregnancy as a marker of cCMV risk [35,36,37,38]. While the risk of a screening test in pregnancy should not be minimized [39], especially in the absence of clinically proven prenatal treatments for cCMV, maternal serologic monitoring may provide an opportunity for targeted neonatal testing and early diagnosis. This is an important area of future research as cCMV screening programs gain traction.

Of those infants tested for cCMV in our study, many had non-specific clinical findings that can also be seen in symptomatic cCMV (e.g., microcephaly, petechiae, small-for-gestational age). As a result, infants with symptomatic disease made up the majority (83%) of those diagnosed with cCMV in the cohort, as opposed to the minority (~10%) as has been found in population-based studies [2,40]. It is likely that provider’s clinical suspicion for cCMV is not raised in the case of asymptomatic infections, due to the absence of visible findings, and therefore these infants are even more likely to go undiagnosed. Of note, although not visible, a failed newborn hearing screen is a finding that clinical providers are made aware of, but this rarely prompted cCMV testing in the study cohort. It may be that some clinical providers lack awareness about the prevalence and presentation of cCMV, as has been suggested by recent work [28]. Infants with cCMV who go undiagnosed likely miss opportunities for early interventions including close auditory follow-up, treatment of hearing loss and developmental surveillance [3,8,41,42,43].

The results from this study highlight the critical need for systematic cCMV screening and testing programs to diagnose most infants with cCMV. Examples of such programs include hearing-targeted cCMV testing, in which all infants who fail their newborn hearing screening are then tested for cCMV, and universal/routine screening programs, in which all infants are screened for cCMV regardless of apparent risk factors [40,44,45]. Such programs have been found to be effective and acceptable in several studies [46,47,48,49,50], although the debate about cost-effectiveness persists [51]. As a more nuanced understanding is being gained about the possible long-term outcomes of cCMV [43,50,52], as well as the effectiveness of early interventions [53,54,55], the risk–benefit analysis may be more clearly weighted towards systematic screening.

This study is not without limitations. In line with the design of a retrospective study, data collection relied on the identification of infants who were tested for CMV and the extraction of clinically significant variables of interest. The identification of clinical indications for testing and clinical findings relied on the documentation by medical providers in the clinical record for the group tested for CMV, which may not accurately or completely capture the physical exam findings. For the study cohort, the provider’s documentation was often vague, allowing for potential selection and misclassification bias, especially for the clinical indication that prompted CMV testing. Furthermore, this study used an electronic query tool to create the cohort, the definitions from which have not been validated. It may be that some infants tested for cCMV were missed. In the larger cohort, ICD-9 and ICD-10 diagnostic codes were used to identify infants with clinical features, which likely underestimates the presence of such signs for the larger cohort. Study results may not be generalizable to other healthcare settings or geographic areas.

## 5. Conclusions

Testing for and the diagnosis of cCMV was infrequent at a US quaternary children’s hospital when testing was reliant on healthcare provider clinical suspicion alone. As most infants with cCMV are born without clinical manifestations, systematic screening programs are critical for early diagnosis and treatment of this common infection.

## Figures and Tables

**Figure 1 IJNS-08-00060-f001:**
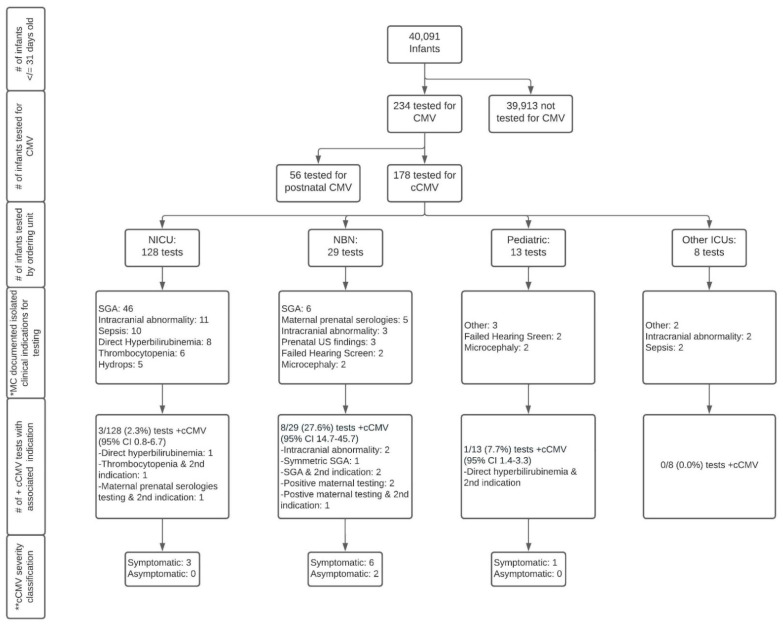
Flowchart of congenital cytomegalovirus testing in infants at a quaternary children’s hospital from 2014 to 2019. Abbreviations: CMV—cytomegalovirus; cCMV—congenital cytomegalovirus; NICU—neonatal intensive care unit; NBN—newborn nursery; Other ICUs—intensive care units including pediatric intensive care unit and pediatric cardiac intensive care unit; MC—most common; SGA—small-for-gestational age; US—ultrasound. Symbols: # number; + positive. * Documented indication by provider; ** Symptomatic cCMV disease classification is based on the International Congenital Cytomegalovirus Recommendations Group definitions as follows [3]: *Symptomatic cCMV*: Clinical manifestations attributed to cCMV including intrauterine growth restriction, petechia, hepatomegaly, splenomegaly, hepatitis (elevated transaminases or bilirubin), thrombocytopenia, or central nervous system involvement including microcephaly, chorioretinitis, detection of CMV DNA in cerebrospinal fluid, abnormal cerebrospinal fluid indices for age, or radiographic abnormalities consistent with cCMV such as ventriculomegaly, intracerebral calcifications, periventricular echogenicity, cortical or cerebellar malformations. May have multiple manifestations and/or CNS involvement (moderate to severe) or 1–2 mild and transient manifestations (mild disease). *Asymptomatic cCMV:* Absence of clinical findings as above. May have sensorineural hearing loss.

**Figure 2 IJNS-08-00060-f002:**
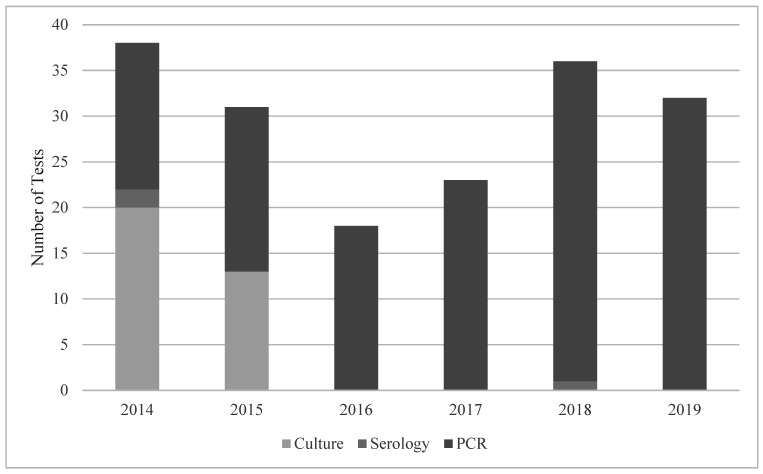
Testing modality for cCMV by year at a quaternary children’s hospital from 2014 to 2019. Abbreviations: PCR—quantitative polymerase chain reaction.

**Figure 3 IJNS-08-00060-f003:**
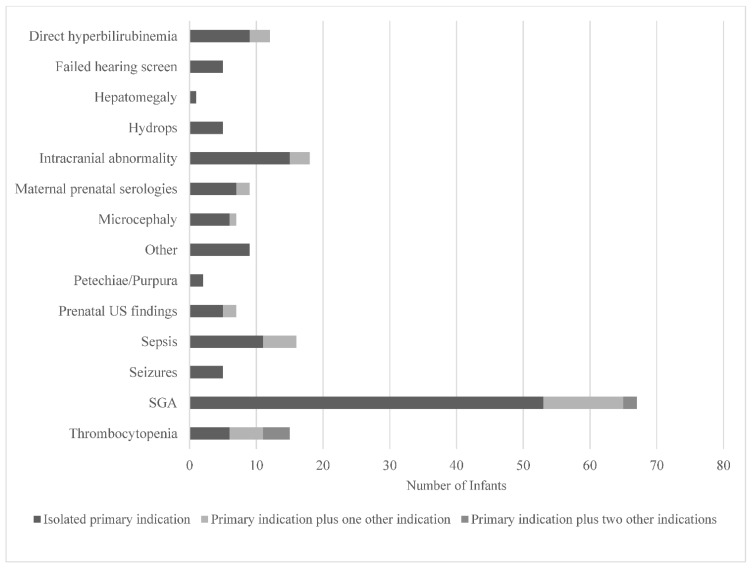
Clinical indications prompting cCMV testing in infants as documented by healthcare providers in a quaternary children’s hospital from 2014 to 2019. Abbreviations: US—ultrasound; SGA—small-for-gestational age. Other includes slow weight gain, intractable diarrhea, cholestasis, liver failure, hemophagocytic lymphohistiocytosis, immunodeficiency, and pre-cardiac transplant evaluation.

**Table 1 IJNS-08-00060-t001:** (**a**,**b**) Characteristics of infants tested for congenital cytomegalovirus at a quaternary children’s hospital from 2014 to 2019 (N = 40,091).

(a) Sociodemographic Characteristics				
	Not Tested for cCMV n = 39,913	Tested for cCMV n = 178	cCMV Negative n = 166	cCMV Positive n = 12
Maternal age (years); mean (SD)	30.2 (5.4) ^a^*	29.8 (5.7) ^a^	29.9 (0.9) ^a^	27.0 (8.0) ^a^
Maternal parity; median (SD)	1.1 (0.3) ^a^*	2 (0.3) ^a^	1(0.3) ^a^	2 (0.8) ^b^
Male sex; n (%)	20,821 (51.9) ^a^	97 (54.5) ^a^	91 (54.8) ^a^	6 (50.0) ^a^
Infant race; n (%)				
African American	4,794 (12.0) ^a^	22 (12.4) ^a^	21 (12.7) ^a^	1(8.3) ^a^
Asian	2,444 (6.1) ^a^	7 (3.0) ^a^	7 (4.2) ^a^	0 (0.0) ^a^
Caucasian	25,621 (64.2) ^a^	119 (66.9) ^a^	109 (65.6) ^a^	10 (83.3) ^a^
Other (e.g., Native American, Pacific Islander, Native Alaskan)	1955 (5.0) ^a^	15 (8.4) ^b^	13 (7.8) ^ab^	1 (8.3) ^ab^
Unknown	5,099 (12.8) ^a^	7 (3.0) ^b^	8 (4.8) ^b^	0 (0.0) ^ab^
Hispanic ethnicity (vs. non-Hispanic or unknown); n (%)	2124 (5.3) ^a^	8 (4.5) ^a^	8 (4.8) ^a^	0 (0.0) ^a^
Public insurance (vs. other); n (%)	15,672 (39.1) ^a^	96 (53.9) ^bc^	90 (54.2) ^bc^	6 (50.0) ^ac^
**(b) Clinical Characteristics**				
	**Not Tested for cCMV** **n = 39,913**	**Tested for cCMV** **n = 178**	**cCMV Negative** **n = 166**	**cCMV Positive** **n = 12**
** *Birth characteristics* **				
Inborn delivery (vs. born at outside hospital); n (%)	31,402 (78.7) ^a^	120 (67.4) ^b^	111 (66.8) ^b^	9 (75.0) ^ab^
Premature < 37^0/7^ weeks; n (%)	4581 (14.6) *^a^	62 (34.8) ^b^	62 (37.3) ^b^	1 (8.3) ^a^
Gestational age (weeks^days^); mean (SD)	38^5/7^ (1^0/7^) *^a^	37^3/7^ (3^1/7^) ^b^	36^1/7^ (4^0/7^) ^c^	37^6/7^ (1^2/7^) ^bc^
Birth weight (grams); mean (SD)	3652.8 (277.2) *^a^	2472.5 (936.1) ^b^	2471.2 (949.9) ^b^	2664.0 (601.5) ^b^
Birth weight %ile for GA; mean (SD)	52.0 (31.1) *^a^	22.5 (32.6) ^b^	31.5 (33.0) ^c^	17.0 (19.2) ^b^
Head circumference (cm); mean (SD)	34.2 (3.7) *^a^	32.8 (3.0) ^b^	32.8 (0.3) ^b^	32.6 (1.8) ^b^
Head circumference %ile for GA; mean (SD)	37.4 (30.0) *^a^	33.0 (32.6) ^ab^	31.5 (37.8) ^b^	18.7 (16.6) ^b^
***Clinical findings/diagnoses* in medical record** (*vs. absent*)*; n (%)*				
Chorioretinitis	2 (<0.0) ^a^	0 (0.0) ^a^	0 (0.0) ^a^	0 (0.0) ^a^
Direct hyperbilirubinemia (>3 mg/dL)	246 (0.6) ^a^	26 (14.6) ^b^	23 (13.9) ^b^	3 (25.0) ^b^
Elevated ALT (>100 IU/L)	547 (1.4) ^a^	30 (16.9) ^b^	27 (16.3) ^b^	3 (25.0) ^b^
Elevated AST (>80 IU/L)	1,174 (2.9) ^a^	56 (31.5) ^b^	48 (28.9) ^b^	8 (66.7) ^b^
Hepatomegaly	74 (0.2) ^a^	24 (13.5) ^c^	19 (11.4) ^b^	5 (41.7) ^b^
Intracranial abnormalities	783 (2.0) ^a^	50 (28.1) ^b^	41 (24.7) ^b^	9 (75.0) ^c^
Jaundice	12,483 (31.1) ^a^	93 (52.2) ^b^	86 (52.0) ^b^	7 (58.3) ^b^
Microcephaly	301 (0.8) ^a^	80 (44.9) ^b^	73 (44.0) ^c^	7 (58.3) ^b^
Petechiae	32 (<0.0) ^a^	20 (12.4) ^b^	15 (9.0) ^b^	5 (41.7) ^c^
Referred/failed hearing screen	2,944 (7.3) ^a^	30 (16.8) ^b^	25 (15.1) ^b^	5 (41.7) ^c^
Seizures	825 (2.1) ^a^	15 (8.4) ^b^	14 (8.4) ^b^	1 (8.3) ^b^
Small-for-gestational age	2,317 (5.8) ^a^	83 (46.6) ^bd^	79 (47.6) ^d^	4 (33.3) ^cd^
Splenomegaly	34 (<0.0) ^a^	11 (6.2) ^b^	7 (2.4) ^b^	4 (33.3) ^c^
Thrombocytopenia (<150 K/uL)	430 (1.1) ^a^	82 (46.1) ^b^	76 (45.7) ^b^	6 (50.0) ^b^

Abbreviations: cCMV—congenital cytomegalovirus; SD—standard deviation; GA—gestational age; %ile—percentile; AST—aspartate aminotransferase; ALT—alanine aminotransferase; ^abc^ Within a row, proportions without a common superscript differ (*p* < 0.01); * N = 31,402 infants inborn with data from birth encounters; SGA and microcephaly were defined as less than the 10th percentile for birth weight and head circumference respectively on the Fenton growth chart for preterm infants (<37 weeks gestation) and the World Health Organization growth chart for term infants (≥37 weeks gestation) [12,13].

## Data Availability

The data presented in this study are available on request from the corresponding author.

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
