# Peer review of "Inconsistent Provider Testing Practices for Congenital Cytomegalovirus: Missed Diagnoses and Missed Opportunities"

_2409-515X, 2022, doi:10.3390/ijns8040060_

Round 1
Reviewer 1 Report
The authors present retrospective descriptive findings for cCMV testing among newborns at a childrens hospital in the U.S.. The authors explore clinical factors which may have prompted testing to obtain serological results. The paper is a unique detailed retrospective descriptive study conducted in the U.S. with results that add to the discussions of the need for routine of focused cCMV testing among newborns. The authors assert that clinical suspicion of cCMV alone provides poor testing and ascertainment of disease, especially for asymptomatic infections.
The paper is well written and thoughtful but needs some work.
Points for clarification:
1) I suggest that the ICCRG cCMV symptoms be enumerated either in the first paragraph where they are first mentioned or elsewhere, such as in the footnote to Figure 2, or in the methods.
2) Introduction: 1st paragraph, 2nd sentence-- The sentence as written suggests that 90% of [all] infants are born with asymptomatic infections, not just 90% of all true cCMV infections which is what the authors want to convey.
3) Methods: Methods should indicate what software was used to calculate p-values, Z-scores, SD and confidence intervals. Also, what were the confidence intervals based on, Z- or t- scores or binomial?
Last sentence typo-- p-value of > 0.05 should be < 0.05.
4) Results: I am not clear why 2 cases transferred to the NICU due to symptoms (of 12 true positive cases) were mentioned in the results. Is that because these 2 had their testing delayed or because they were initially identified by another institution and transferred? If it is the later then inclusion indicates that symptomatic identification of cCMV in the NICU is extremely poor (0.008, 1/126) not (0.02, 3/128), and this should be mentioned in this paragraph. I don't suggest changing the tables or the flowchart, instead clarify the implication of this inclusion. However, if these transferred cases were initially identified from another institution in a regular nursery or after release (pediatric?) they may belong among the NBU in the flowchart instead.
5) Discussion: 2nd paragraph-- The authors can include a sentence about poor ascertainment in the NICU, where SGA is more common and not predictive of cCMV. The authors could use the data from figure 1 to make a few demonstrative calculations ex. NICU, 46 SGA/128=36% and no (0?) positive results related to SGA, vs NBU 6 SGA/29=21% and 3/6 testing and diagnosis were partially prompted by SGA. This observation can emphasis the thrust of this article.
6) Table 1b-- Elevated ALT and AST are missing statistical test results.
Reviewer 2 Report
General comments
In the absence of screening at birth few infants are tested for cCMV, mostly those with clinical signs, and few infants with cCMV infections are clinically diagnosed. Leung et al. (J Pediatrics 2022) summarized what is known, “The administrative prevalence of cCMV is typically more than 10 times lower than the prevalence of cCMV identified through newborn screening. Most, but not all, infants with cCMV diagnoses have symptomatic cCMV.” It is not clear that another study is needed to demonstrate either those well-established facts or the implication that current clinical practice results in numerous missed diagnoses and missed opportunities for treatment.
The authors use the presence of diagnosis codes in EHRs to identify presumed signs that prompted testing for cCMV. However, the presence of a sign does not by itself indicate why testing was performed. Previous analyses of administrative data have documented the presence of diagnoses among infants who either undergo testing or are diagnosed with cCMV. For example, Leung et al. (2013) reported that more than 80% of infants with procedure codes for CMV tests had one or more diagnoses of clinical signs of cCMV. The authors might have made an important contribution by accessing the clinical data contained in EHRs to identify the actual reasons why laboratory tests were ordered. For example, one previous study from Montreal examined medical records, revealing that 51% of infants diagnosed with cCMV were tested because of disease symptoms, 34% were tested because of suspected maternal infection during pregnancy, 10% were tested because they did not pass newborn hearing screening, and 4% were tested because of maternal HIV (Smilkjovic et al. 2020).
Specific comments organized by section
Introduction
The authors cite two estimates of the frequency of clinical diagnoses of cCMV in routine practice without NBS for CMV. First, they cite reference 6 as reporting that the number of infants diagnosed with cCMV in British Columbia was equivalent to 1% of the true prevalence of cCMV and roughly 10% of the prevalence of symptomatic cCMV. Second, reference 7, indicated that roughly 2 per 10,000 infants are typically diagnosed with cCMV, which is equivalent to roughly 5% of all infants with cCMV (4-5 per 1,000) and 40-50% of those with symptomatic cCMV. The authors should acknowledge that those estimates have very different implications regarding the proportion of symptomatic cases that are diagnosed in the absence of screening.
Methods
The authors report the frequency of 6 selected procedure codes for laboratory tests ordered in infants of 31 days of age or less. CPT codes 87497, 87496, 87252, 87254, 86644, and 86645 were utilized to “identify all infants tested for any type of CMV.” Unfortunately, the authors appear to have limited familiarity with laboratory testing for CMV. Although the first two and the last two are CMV-specific test codes, the other two are not specific to CMV. Because 87252 and 87254 can be used to test for a large number of different viruses, it is incorrect to refer to those as codes associated with CMV testing.
Among previous studies of the frequency of documentation of laboratory testing for CMV in infants, Leung et al. (2013) examined insurance claims data from 2011 for CPT codes to identify newborns who might have been tested for CMV. They used 8 CPT codes that are specific to CMV and another 17 non-specific CPT codes for viral testing that can be used to test for either CMV or other viruses. The authors should either explain why they do not consider the other CMV-specific CPT codes used by Leung to be appropriate for their own study or repeat their analysis incorporating those procedure codes. In addition, they should report how many infants were tested for each specific type of laboratory test. In particular, CMV serology testing (codes 86644 and 86645) is not recommended to test for cCMV because of its low accuracy, although as of 2011 its use was still common as noted by Leung et al. How common was serology testing in the study sample? Did that decrease over time, with increasing use of PCR tests?
The use of non-specific test procedure codes requires more explanation. Unless the authors have additional information that they did not report demonstrating that infants were specifically tested for cCMV using those codes, it is misleading to imply that the infants had been tested for CMV.
Campione et al. (2022) examined an EMR database for records of positive CMV laboratory tests in newborn infants during 2010 to 2017. Unfortunately, the authors of that study did not report how many tests were performed or how they were identified, whether through CPT codes or LOINC codes. Also, although they considered PCR, direct fluorescent antibody [DFA], culture or IgM serology, the vast majority of tests involved culture. Campione et al. acknowledged that they likely missed PCR tests due to poor recording. Curiously, only about 10% of infants who were confirmed as having cCMV through a positive viral culture test had a cCMV diagnosis code in their EMR. The conclusion was that many more infants are being diagnosed with cCMV than are receiving cCMV diagnosis codes. The authors of the present submission should discuss this argument and its implications for their own data.
The flowchart in Figure 1 is insufficiently documented. It indicates that 234 infants were tested for CMV, of whom 178 were tested for cCMV and 56 for postnatal CMV. How did the authors make that determination? There is no mention in the text of the operational definitions used in generating those estimates in Figure 1.
The authors state that diagnoses of cCMV were based on documented CMV positive PCR or culture tests in the first 21 days of life or with a confirmatory dried blood spot PCR after 21 days. It is not stated how many of those infants had cCMV diagnosis codes recorded. It is also unclear how many infants in their study population had cCMV diagnosis codes recorded but without documentation of a positive CMV lab test. Campione et al. reported that most infants who had a positive CMV test result, primarily in cultures, did not have a diagnosis code recorded. How often did the same occur in the present study?
Results
The authors repeatedly refer to “indications documented by healthcare providers that prompted testing.” The Methods section does not make any mention of documentation of indications for testing. Instead, it refers to “circumstances around cCMV testing, sociodemographic and clinical characteristics of the infants”. That information is at best circumstantial evidence of possible reasons for ordering of tests. Moreover, infants may have had multiple clinical signs or reasons for ordering testing. The fact that the most common “sign” was SGA does not mean that SGA was the indication for testing. It is misleading to confuse potential indications with actual indications for testing. It would have been more informative if the authors had been able to document the reported indications for the ordering of tests.
The authors report useful information from the clinical data contained in EHRs for the 178 infants who were either tested for cCMV or some other virus. For example, they reported that 17% of infants who were tested had failed a newborn hearing screen and 42% of those who tested positive had failed hearing screening. It would be even more informative if the authors were able to report in how many infants the failed hearing screen was the actual indication for testing rather than just one of multiple “indications”. Doing so would allow them to compare their single-center study with the findings previously reported from a children’s hospital in Montreal (Smilkjovic et al. 2020).
It is also very interesting that microcephaly was present in 45% of infants tested and 58% of those positive for cCMV. Those proportions may be higher than reported in studies that relied on the presence of diagnosis codes as opposed to clinical records. It could be informative if the authors were to compare the prevalence of clinical signs in their data with previous published reports that used administrative data.
Discussion
The discussion of indications for testing is not adequate. The authors appear to have confused the presence of clinical signs such as SGA with indications for testing. If the authors obtained actual information on indications for testing, that needs to be documented in the Methods section.
The most intriguing finding mentioned in the Discussion section is “positive maternal prenatal CMV serologies was the most common indication that was associated with a positive result”. It is reported that the mothers of 4 of 12 infants who tested positive had positive prenatal serologic tests. That suggests that prenatal testing may be a very important indication for testing newborns for cCMV, which is consistent with previous findings from Montreal (Smilkjovic et al. 2020). The authors should clearly document the information on prenatal serologic test results in the Methods and Results sections prior to addressing it in the Discussion. All findings that are discussed in the Discussion section of a paper should first be presented in the Results section as well as have clear documentation in the Methods section.
The authors report that 2-3 per 10,000 infants at their institution were diagnosed with cCMV, which they state is consistent with Canadian and UK studies, citing references 6 and 13. Reference 6 reported a prevalence of 0.4 per 10,000 infants in British Columbia and reference 13 reported a prevalence of 0.6 per 10,000 in the British Isles (UK and Republic of Ireland), both of which are much lower, In contrast, the findings of the submission are fully consistent with the broader literature reviewed in reference 7.
Round 2
Reviewer 2 Report
This is one of the most impressive and thorough revisions of a peer-reviewed manuscripts that I have ever seen. I am in awe of the work done by the authors. This paper represents an important contribution to the field of research on the diagnosis of congenital CMV under routine clinical practice using real-world data.
The authors were very responsive to all of my comments with one notable exception. My original comment was, “Second, reference 7, indicated that roughly 2 per 10,000 infants are typically diagnosed with cCMV, which is equivalent to roughly 5% of all infants with cCMV (4-5 per 1,000) and 40-50% of those with symptomatic cCMV. The authors should acknowledge that those estimates have very different implications regarding the proportion of symptomatic cases that are diagnosed in the absence of screening.”
The authors responded that according to their calculations “the US administrative claims study are comparable, with ~10% of all symptomatic cases or 1% of all cases of cCMV identified.” However, the authors neither reported their own calculations nor identified any flaw in the calculations that I had reported. In my calculations I assumed a birth prevalence of 4-5 per 1,000 births. The authors prefer a broader range of 3-6 per 1,000 births. With roughly 10% of all cCMV cases symptomatic at birth, the authors’ estimate of birth prevalence of 3-6 per 1000 births implies between 3 and 6 per 10,000 U.S. births have symptomatic cCMV. A diagnosed rate of 2 per 10,000, as reported in reference 7 represents between 1/3 and 2/3 of symptomatic cCMV cases, not 10%. Consequently, the author's conclusion that very few infants with symptomatic cCMV are being diagnosed in the absence of newborn screening is not supported by the evidence presented.
Round 3
Reviewer 2 Report
The authors have been fully responsive. I think the edits may have gone further than necessary and detract from the authors' arguments.
L51: Replace "10x" with "5x"
L52-53: Suggest deletion of sentence beginning, "There may be...."
L55-56: Suggest substitution of "dramatic" for "beneficial"
L56-57: Suggest deletion of sentence beginning, "Especially..."